# Chitosan/Alginate Hydrogel Dressing Loaded FGF/VE-Cadherin to Accelerate Full-Thickness Skin Regeneration and More Normal Skin Repairs

**DOI:** 10.3390/ijms23031249

**Published:** 2022-01-23

**Authors:** Lai Wei, Jianying Tan, Li Li, Huanran Wang, Sainan Liu, Junying Chen, Yajun Weng, Tao Liu

**Affiliations:** Key Laboratory of Advanced Technology of Materials, Ministry of Education, Southwest Jiaotong University, Chengdu 610031, China; laiwei_myfuture@163.com (L.W.); jianying1128@126.com (J.T.); lili1993@my.swjtu.edu.cn (L.L.); huanran@my.swjtu.edu.cn (H.W.); liusainan1995@163.com (S.L.); wengyj7032@swjtu.edu.cn (Y.W.); lt045021@gzucm.edu.cn (T.L.)

**Keywords:** hydrogel dressing, full-thickness skin regeneration, 3D cell culture, FGF, VE-cadherin

## Abstract

The process of full-thickness skin regeneration is complex and has many parameters involved, which makes it difficult to use a single dressing to meet the various requirements of the complete regeneration at the same time. Therefore, developing hydrogel dressings with multifunction, including tunable rheological properties and aperture, hemostatic, antibacterial and super cytocompatibility, is a desirable candidate in wound healing. In this study, a series of complex hydrogels were developed via the hydrogen bond and covalent bond between chitosan (CS) and alginate (SA). These hydrogels exhibited suitable pore size and tunable rheological properties for cell adhesion. Chitosan endowed hemostatic, antibacterial properties and great cytocompatibility and thus solved two primary problems in the early stage of the wound healing process. Moreover, the sustained cytocompatibility of the hydrogels was further investigated after adding FGF and VE-cadherin via the co-culture of L929 and EC for 12 days. The confocal 3D fluorescent images showed that the cells were spherical and tended to form multicellular spheroids, which distributed in about 40–60 μm thick hydrogels. Furthermore, the hydrogel dressings significantly accelerate defected skin turn to normal skin with proper epithelial thickness and new blood vessels and hair follicles through the histological analysis of in vivo wound healing. The findings mentioned above demonstrated that the CS/SA hydrogels with growth factors have great potential as multifunctional hydrogel dressings for full-thickness skin regeneration incorporated with hemostatic, antibacterial, sustained cytocompatibility for 3D cell culture and normal skin repairing.

## 1. Introduction

In the human body, the skin is the most extensive and most vulnerable tissue. Skin also plays a significant role in defending external damage and microbial infection [1]. Once the skin tissue is damaged, the cutaneous wound healing might be a complex multi-step process that involves related dermal and epidermal events. During the skin repairing process, lots of soluble factors, blood elements, extracellular matrix (ECM) and cells were involved [2]. The skin repair may generally be divided into four continuous phases: hemostasis, inflammation, proliferation and remodeling [3]. Although most incisional skin wounds can be effectively healed, excisional wounds, also called the extensive full-thickness wound usually hard to repair, causing severe infecting and thus threatening life. Thus, the design of wound dressings with multifunctional properties is highly desired.

The functions of wound dressings may include but are not limited to covering the wound and acting as a temporary barrier, guiding the reorganization of skin cells and re-integration of host wound skin tissues, including epidermis, basement membrane and dermis. An ideal skin wound dressing should meet the requirements below: sufficient mechanical strength, good moisture retention, appropriate surface microstructure, excellent tissue compatibility [4]. A variety of biomedical materials can be served as wound dressings, such as fibrous [5] membranes [6], polymer and polysaccharide scaffolds [7], nanoparticles [8], and hydrogels [9]. Compared with other dressing materials, hydrogels dressings [10,11] are attracting increasing attention due to their adjustable physicochemical properties, similar to ECM, able to modulate the fluid balance and accelerate wound repair [12,13]. Several hydrogel dressings with remarkable antibacterial activity and excellent promotion to wound healing [14,15]. Thus, developing a multifunctional hybrid hydrogel dressing to promote the full-thickness wound healing process is highly desirable.

Chitosan (CS) [16,17] and alginates (SA) [18,19], as natural polysaccharides, have received increasing attention due to their high hydrophilicity and outstanding biocompatibility. CS [20,21], as the unique cationic polysaccharide, shows high antimicrobial activity due to the interaction between positively charged CS and the negatively charged bacterial membrane [22]. Among these materials, CS may be considered as an ideal candidate in the wound repair stages of inflammation [23]. Besides, due to the outstanding biocompatibility, suitable biodegradability and low toxicity, CS has become one of the most extensively used hydrogel dressings [24]. In addition, CS can induce local macrophage proliferation, stimulate the remodeling of ECM and thus promote early wound healing. Furthermore, CS processes abundant primary amine groups, which endows them to react with carboxyl groups [25,26,27]. SA, a natural anionic polysaccharide (carboxyl groups), is of particular interest for skin dressings due to their non-toxic and biocompatibility [28]. Concerning their chemical formula and structure, SA own ease of gelation kinetics and similarity to natural ECM [29], which makes them proper candidates compared to other biomaterials. Although SA lacks an adhesive motif that can initiate cell adhesion, the addition of growth factors may help improve the cell adhesion.

Growth factors belonging to the fibroblast growth factors (FGF) family play crucial roles in tissue repair and regeneration. The FGF family are small proteins and process a typical β-barrel structure as the core [30]. The FGF can be divided into three major groups: canonical, hormone-like, and intracellular. The FGF plays a vitally important role in tissue repair processes after mechanical injury, burns and chemical damage [31]. In general, the function of FGF includes stimulating cell proliferation and migration [32]. FGF could also enhancing angiogenesis which promote the formation of new vessels from pre-existing vessels [33,34]. FGFs can regulate many aspects of the cell phenotype, which is critical to tissue repair. For example, for endothelial cells, FGF2 could promote lifespan extension and angiogenesis, suppress apoptosis [35]. For fibroblasts, FGF2 could suppress their differentiation to myofibroblasts. Vascular endothelial cadherin (VE-cadherin) is an endothelial-specific marker expressed by endothelial cells during vasculogenesis [36] and a major component of endothelial adherens junctions. VE-cadherin is crucial for the organization of the newborn vascular network [37]. Besides, the function of VE-cadherin also includes promoting endothelial differentiation and mediating the interstitial mechanotransduction [38].

In this study, taking advantage of the hydrogen bond and covalent bond between chitosan and alginate, a series of complex hydrogels were synthesized. Furthermore, poly (ethylene glycol) diacrylate (PEGDA) was introduced to increase the mechanical property of hydrogels. These hydrogels exhibited high swelling properties, suitable degradation, proper aperture for cell growth and tunable rheological properties. Besides, the performance of hemostatic ability, antibacterial behaviors both in vitro and in vivo, cytocompatibility for L929 (epithelioid fibroblasts cells) and EC (endothelial cells) were systematically investigated. The super hemostatic, antibacterial and cytocompatibility in vitro confirmed the biocompatibility of hydrogels. Furthermore, wound closure, epidermal thickness and histological examinations were carried out to evaluate the effects of full-thickness wound healing. In a word, all the results demonstrated that these complex hydrogels possess fantastic ability for the full-thickness skin defected repairing process, which is involved in antibacterial, proper hemostatic, cell proliferation and tissue regeneration.

## 2. Results

### 2.1. Physical Characterization of Complex Hydrogels

The structural formula and schematic drawing of the CS/SA/PEGDA complex hydrogels are showed in Figure 1. The volume ratio of mixing solution and SA were 6:4, 7:3, 8:2, and were named C6-S4, C7-S3, C8-S2, respectively. After that, VE-cadherin (20 μg/mL) and FGF (100 ng/mL) were added to the C8-S2 hydrogel and signed as C8-S2-V and C8-S2-F, respectively. The hydrogel added to the VE-cadherin (20 μg/mL) and FGF (100 ng/mL) at the same time were signed as C8-S2-F-V.

The complex hydrogels were semi-transparent and non-flowing condition. The proper water uptake ability of hydrogel helps to accelerate the process of wound healing through absorbing exudate from wounds and reducing the risk of infection, which was one of the requirements for wound dressings. Swelling tests were used to determine the swelling ratio (SR) of the complex hydrogels, which were analyzed by the weight ratio of dried state (Wd) and fully swollen state (Ws). The equilibrium mass swelling of complex hydrogel was calculated according to the formula: SR = (Ws − Wd)/Wd. As showed in Figure 2a, the swelling ratio of C6-S4, C7-S3 and C8-S2 hydrogels were 4.87, 5.32 and 8.86, respectively. The equilibrium mass swelling of hydrogel increases with the increase of the amount of chitosan in the hydrogel.

Figure 2b results showed scanning electron microscope (SEM) micrographs of the cross section of the complex hydrogels. The pores present elliptical shapes, and the pore size changed with the different proportions of hydrogels. In general, with the increase of chitosan content, the major axis of elliptical shapes increased. According to the calculation, C6-S4 showed the smallest pore size (about 32 μm for minor axis and 64 μm for major axis), and C8-S2 showed the largest pore size (about 38 μm for minor axis and 105 μm for major axis). Pore size at dozens of micrometers is beneficial to the transportation of nutrients and convenient for the ingrowth of cells. The different aperture sizes were mainly due to the different crosslinking degrees of the hydrogel, which suggested the crosslinking degree decreased with the increase of chitosan content.

To analyze the influence of different chitosan content on the rheological properties of the complex hydrogels, the curves of storage modulus (G′) and loss modulus (G″) of the hydrogels over frequency were recorded. The G′ of C6-S4 hydrogel (5848 Pa) was higher than C7-S3 (4428 Pa) hydrogel and C8-S2 (2778 Pa) hydrogel at 100 Hz. The rheological properties showed the same rule at low frequency. The G′ of C6-S4 hydrogel (764 Pa) was higher than C7-S3 (302 Pa) hydrogel and C8-S2 (211 Pa) hydrogel at 10 Hz. This is because more chitosan content in the hydrogel decreased the crosslinking density and thus leading to the lower G′ of the hydrogel (Figure 2c).

The fourier transform infrared spectroscopy (FTIR) of alginate (SA), chitosan (CS) and C6-S4 hydrogel were showed in Figure 2d. The characteristic peak of hydrogen bond may be found at 3422 cm^−1^, The characteristic peak of amido bond and carbon and oxygen double bond may be found at 1598 cm^−1^ and 1620 cm^−1^, respectively. The characteristic peak of carbon and nitrogen double bond may be found at 1637 cm^−1^. 

After immersing in PBS for 11 days, the hydrogels condition was taken photos (Figure 2e). For the C6-S4 sample, the hydrogel was almost complete and maintained a non-flowing state. In contrast, for C7-S3 and C8-S2 samples, the hydrogel resolved in PBS with large pieces of gels and represented a flowing state. Degradation tests were used to determine the Degradation ratio (DR) of the complex hydrogels. Firstly, the complex hydrogels were immersed in 0.01 M PBS at 37 °C with shaking at 70 rpm until the weight of all hydrogels was kept constant and the weight were denoted as W_0_. At the predetermined time, the hydrogels were taken out, rinsed using RO water to remove excess salinity, the superficial water of gels were removed by filter paper, and the weight was denoted as W′. DR of hydrogels was analyzed by the formula: DR = (W_0_ − W′)/W_0_ × 100%. According to the calculation, for C6-S4, C7-S3 and C8-S2 hydrogel, the mass loss was 15.18%, 24.99% and 42.81% after degradation for 5 days, respectively. Similarly, for C6-S4, C7-S3 and C8-S2 hydrogel, the mass loss was 27.40%, 64.88% and 73.73% after degradation for 11 days, respectively (Figure 2e).

Figure 2f showed SEM micrographs of the cross section of the complex hydrogels after degradation for 11 days. Compared to the SEM micrographs before and after degradation, the pore size became larger, and communicating pores appeared. In general, after degradation for11 days, these hydrogels still maintained hydrogel condition and processed certain crosslinking degrees. According to the SEM, with the increase of chitosan content, the pore size of the hydrogel increased and the connection between holes thinner.

### 2.2. Hemolysis and Whole Blood Dynamic Coagulation Evaluation Results

As showed in Figure 3a, the supernatant is transparent and without apparent hemolysis. Furthermore, the hemolysis ratio of all the hydrogels were less than 5% (Figure 3b), which accorded with the standard of blood contacted material. When the blood is connected to the hydrogels, the connection may lead to the rupture of red blood cells and cause hemolysis. Therefore, the hemolysis test is an essential experiment for the safety of blood connected materials.

The whole blood dynamic coagulation assay was carried out to detect the coagulation after the blood was connected to the hydrogels. After the hydrogel was connected to the blood for 10, 20, 30, 40 and 50 min, the blood clots that coagulated on hydrogels were showed in Figure 3c. For C6-S4 hydrogels, there were hardly any blood clots within 20 min, only a tiny amount of blood clots was formed during 20–30 min, and mass of blood clots began to form after 30 min. These results implied that the formation of large blood clots was postponed to 30 min after the hydrogel connected to the blood for C6-S4. In contrast, for C8-S2 hydrogel, there were few blood clots after the hydrogel connected to the blood for 10 min. After that, more and more blood clots began to form, which means the significant coagulation reaction occurred after 10 min connection. Figure 3d showed the absorbance of hydrogels at different point times. As time went by, more and more blood involved in the coagulation reaction; the uncoagulated blood became less and led to a lower absorbance. Therefore, the lower the absorbance, the better the coagulation effect. The blood coagulant index (BCI) of hydrogels were analyzed by the following formula: As/Aw × 100%. After the connection of 10 min, the BCI of all hydrogels were above 40%, which exhibited soft coagulation and prevented the blockage of capillaries (Figure 3e). Ten minutes later, the C8-S2 hydrogels began to form blood clots to prevent the hemorrhage of large vessels.

### 2.3. Antibacterial Activity Assessment

The antibacterial activities of complex hydrogels against Gram-negative bacteria *E. coli* and Gram-positive bacteria *S. aureus* were conducted by surface antibacterial assay, and LB agar gel plates without hydrogels were set as control and marked as blank. After incubation at 37 °C for 12 h, the hydrogel plates and agar gel plates were taken photos to observe the colony-forming units (CFUs) on each plate. There were no CFUs on every plate (Figure 4a,d). Then, in order to detect the bacteria on each plate, 1 mL sterilized PBS was added to each hydrogel plate to dissolve survived bacteria and then the suspension was spaced on the agar gel surfaces and incubated. The killing efficiency was calculated using the following formula: (N_1_ − N_2_)/N_1_ × 100%, where N_1_ refers to the number of CFUs of control, and N_2_ refers to the number of CFUs of the survive to count on complex hydrogels. At the bacterial concentrations of 10^3^ and 10^5^ CFU/mL, the CFUs were not found on the complex hydrogels, and a large amount of CFUs appeared on the control plate (agar gel), exhibiting 100% killing efficiency against *E. coli*. At the bacterial concentrations of 10^2^ and 10^4^ CFU/mL, the CFUs were not found on the complex hydrogels, and a large amount of CFUs appeared on the agar gels, exhibiting 100% killing efficiency against *S. aureus*. When the bacterial concentration was increased to 10^6^ CFU/mL, the hydrogels also exhibited high killing efficiency against both microorganisms. The killing efficiency of C6-S4, C7-S3 and, C8-S2 hydrogel against *E. coli* was up to 97.1%, 100% and 100%, respectively (Figure 4b). Similarly, the killing efficiency of C6-S4, C7-S3 and, C8-S2 hydrogel against *S. aureus* were up to 96.5%, 100% and 100%, respectively (Figure 4e). The process of wound healing may be delayed or cause infection due to bacterial infection. Therefore, the antibacterial assessment aiming to detect the antibacterial ability of hydrogel as wound dressings are of vital importance. So, as an ideal wound dressing, the antibacterial ability should become their inherent property which could reduce the complications and accelerate the healing process of the wound site.

The in vivo antibacterial activities of complex hydrogels against *E. coli* and *S. aureus* were evaluated via rats’ full-thickness infected skin defect model. After adding the bacterial suspension to the defected skin for 30 min, hydrogel dressing was covered to the defected skin for 24 h, and the wound tissues were harvested and stained by the gram. The tissue containing *E. coli* could appear red pattern, the tissue containing *S.aureus* could appear bluish violet pattern, and the healthy tissue could appear pink pattern. The more red or bluish violet patterns, the more *E. coli* or *S. aureus* bacteria. For *E. coli* groups, no red pattern appeared on all the hydrogel groups, suggesting the excellent antibacterial property against *E. coli* (Figure 4c). For *S. aureus* groups, there were little bluish violet patterns on the C6-S4 hydrogel group, and there were scarcely any bluish violet patterns on the C8-S2 hydrogel group (Figure 4f). Therefore, C8-S2 hydrogel showed better antibacterial properties than the C6-S4 hydrogel group against *S. aureus*.

### 2.4. Three-Dimensional Encapsulation of Cells in Complex Hydrogels

To further evaluate the cytocompatibility of hydrogels, the L929 cells and ECs were encapsulated into the hydrogels. The morphology of adherent cells was stained by rhodamine 123, and the number of cells was detected by CCK-8 assay. As showed in Figure 5a, the morphology of endothelial cells on a tissue culture plate (TCP) showed a typical cobblestone shape morphology. However, the morphology of endothelial cells on hydrogels (C6-S4, C7-S3 and C8-S2) showed a spheroidal morphology and un-spread state (Figure 5a). In terms of cell viability, all the hydrogel groups showed no apparent difference within five days (Figure 5b). However, in terms of the proliferation of cells, the C8-S2 hydrogel revealed better proliferation than the C6-S4 hydrogel (Figure 5c). The morphology of L929 in hydrogels was also analogous with the EC, which exhibited spherical and non-spread shape (Figure 5d). For the culture of L929, the C8-S2 hydrogel exhibited a significant difference with the C6-S4 hydrogel in the aspects of cell viability and proliferation (Figure 5e,f), indicating the promoting effect of chitosan on cell proliferation.

### 2.5. Three-Dimensional Encapsulation of Two Factors and Two Cells in Complex Hydrogels

To further evaluate the cytocompatibility of these hydrogels, the co-culture of L929 and EC with the incorporation of FGF and VE-cadherin were carried out. As showed in Figure 6a, there were only a few cells on all the hydrogel samples within three days. The cell viability of all hydrogel groups on the 1st and 3rd days showed no noticeable difference (Figure 6b). However, the number of cells on the 7th day significantly increased compared to the 3rd day, demonstrating that the cells grew well and were in a state of proliferation. After 12 days’ co-incubation, C8-S2-F and C8-S2-F-V hydrogels exhibited higher cell number and better cell proliferation than other hydrogels, indicated the promoting effect of FGF on cells proliferation (Figure 6c). The above experiments confirmed the brilliant cytocompatibility on the co-culture of L929 and EC. Generally speaking, cells tend to form multicellular spheroids (Figure 6d). At the same time, the transportation of oxygen and nutrients to the core may be difficult when the spheroids grow. Therefore, the spheroids’ diameter is limited to 200–400 μm [39].

### 2.6. In Vivo Repair and Regeneration Evaluation

To further evaluate the healing process of wound skin, a full-thickness infected skin-defect *rat* model (Figure 7a) were prepared, and histological analysis was performed. As time went by, the wound areas gradually became smaller (Figure 7b). Specifically, for groups hydrogel with factors (C8-S2-V, C8-S2-F and C8-S2-F-V), the wound areas were smaller than the primary hydrogel group (C8-S2) at 3, 7 and 14 days after surgery. Therefore, the percentage of wound closure of hydrogel with factors was higher than the pure hydrogel group. Additionally, compared to the NaCl group, the groups of hydrogels with factors showed complete and more appropriate epithelial thickness on the 14th day (Figure 7c). The epidermal thickness of hydrogel with factors was closer to normal healthy tissue (70–120 μm). For further insight into wound regeneration, hematoxylin eosin (H&E) staining was performed to analyze the changes of epidermal, dermis, skin appendages and new blood vessels. By monitoring the wound site in the whole stages of wound healing (3–14 d), the inflammatory response reduced and no obvious inflammatory response for all the groups on the 7th day after surgery (Figure 7d). New blood vessels are crucial to wound healing by providing nutrients to the damaged sites. In general, new blood vessels and skin appendages like hair follicles were observed in all the groups. On day 14, the group for hydrogel with factors showed better-organized granulation tissue at wound microenvironment. As showed in Figure 7d, hydrogel groups exhibited more blood vessels and hair follicles than the NaCl group on the 14th day. These results indicated that the hydrogel groups, especially for C8-S2-F and C8-S2-F-V groups, were beneficial to extracellular matrix remodeling and tissue regeneration. Hydrogels with stereoscopic structure provide cells with three-dimensional growth, promote the interaction between biomolecules, cells and tissues and thus enhance the wound healing process. In addition, the micro-pores in the hydrogels make it easy for cells and nutrients to transportation and communication.

## 3. Discussion

### 3.1. The Influence of Chitosan on the Biological Properties of the Complex Hydrogels

The formation of complex hydrogels was showed in Figure 1. There were mainly three types of chemical bonds in the hydrogels: (1) Hydrogen bond between the hydroxy of chitosan and the carboxyl of alginate, which is the main bonding in the hydrogel. This bonding can support the formation of hydrogel independently without the adding of EDC/NHS. 1-3-Dimethylaminopropyl-3-ethylarbidiimide hydrochloride (EDC, C8H17N3·HCl) and N-hydroxysuccinimide (NHS) are often used as binding agents of the amino and carboxyl groups. (2) The amido bond between the amino of chitosan and the carboxyl of alginate, whose formation had to rely on the adding of EDC/NHS. (3) Carbon and nitrogen double bond between carbon and oxygen double bond of PEGDA and the amino of chitosan, which could increase the stability of the hydrogel. Additionally, the increased concentration of PEGDA could enhance the rheological properties of the hydrogels. Moreover, with the introduction of NaHCO_3_, the pore size of the hydrogel is adjustable through the concentration of NaHCO_3_. In general, through the controlled ratio and concentration of these three materials, the hydrogels could be adjustable in terms of the pore size, degradation rate and rheological properties and thus further influence the biological properties of the hydrogels.

Concerning the coagulation affect and better cytocompatibility, C8-S2 hydrogels were better than C6-S4 hydrogels. All the hydrogels own the same excellent antibacterial effect. Therefore, the increase of chitosan could promote the coagulation, antibacterial and adhesion of cells. Firstly, the hemostatic effects of chitosan took place mainly via the following three mechanisms: (1) Stimulation of platelets. There were lots of literatures have showed that chitosan could induce platelet adhesion and aggregation [40]. (2) Aggregation of red blood cells. Recently, more evidence showed that the hemostatic promotion of chitosan was independent of traditional clotting pathways [41,42]. Due to the opposite charges of chitosan and red blood cells, chitosan attracted red blood cells and formed a “mucoadhesive barrier” via direct interaction at the wound site to stop the bleeding [43]. (3) Alteration of the structure of fibrinogen. The electrostatic forces of ionized chitosan may result in changes in the structure and function of fibrinogen. Secondly, the antibacterial mechanism of chitosan may be explained by the two following [44]: (1) Permeabilization. Chitosan could cause the permeabilization of bacterial via the electrostatic interaction between chitosan and bacteria. The permeabilization may cause the leakage of intracellular substances and thus lead to the death of bacterial. (2) Binding with the nucleic acid. Some research showed that chitosan could affect the DNA expression via binding with the nucleic acid of bacteria. Besides, many factors could affect the antibacterial of chitosan, such as the deacetylation degree and concentration of chitosan, pH value, temperature and chitosan derivatives, etc.

### 3.2. The Influence of FGF and VE-Cadherin in 3D Hydrogels on the Cell Proliferation and Tissue Regeneration

Cells used to be cultured in two-dimensional plates, which have been proved to distort the behaviors and functions of cells compared to the cells in vivo. In 2D cells culture plates, cells usually cannot establish sufficiently efficient cell to cell links and cell to extracellular matrix (ECM) links, which may induce the changes of cells’ morphology and gene expression. Therefore, to obtain the same morphology and functions as cells in vivo, the design of a three-dimensional cell culture niche was introduced to mimic the microenvironment in vivo, which could provide sufficient cell to cell links and interactions [10]. However, the difficulties of 3D cell culture are the supplement of oxygen and nutrient for the inner cells, which has become a bottleneck of the hydrogel application. An ideal 3D cell culture niche ought to possess excellent biocompatibility for cells proliferation and appropriate porous structure for the transportation of oxygen and nutrient. According to the results mentioned above, the C8-S2 hydrogels could be designed as a great 3D cell culture niche. With the adding of FGF and VE-cadherin, this 3D cell culture niche became more excellent. Generally, FGF and VE-cadherin both could induce the adhesion, proliferation and migration of cells through the following signaling pathways showed in Figure 8.

The combination of FGF and FGFR may result in phosphorylation of FGF and activate the following signaling pathways [25]: the binding of FGFR to the adaptor protein (CRKL) may result in the bonding of FRS2α, and thus leads to the adaptor protein (GRB2 and GAB1). GRB2 could activate the Ras-MAPK signaling pathway. At the same time, GAB1 could activate the PI3K-AKT signaling pathway. The interactors of VE-cadherin and stabilized endothelial adherens junctions were showed in Figure 8. The interaction of α-catenin with VE-cadherin needs to via β-catenin, which is dynamically contacted to the cytoskeleton, thus activating F-actin and PI3K/AKT pathway. In stabilized endothelial adherens junctions, the activity of the PI3K/AKT pathway are induced in two ways; one is the phosphorylation of 120, 14-3-3 complex and YAP, the other is CCM1, CCM2 and CCM3 [45]. VE-cadherin can also form complex with many different transmembrane signaling systems, such as the vascular endothelial growth factor receptor-2 and the FGFR1, etc. These complexes are essential in terms of the regulation of the activities of the cell-to- cell junctions.

Besides the adhesion of cells, angiogenesis also plays an essential role in skin wound healing [46]. There was evidence that proved that in VE-cadherin-deficient embryos, there was no sign of angiogenic sprouting. Furthermore, VE-cadherin activities have been proved to participate in the process of blood vessel proliferation and migration [47].

## 4. Experimental and Methods

### 4.1. Materials and Reagents

Chitosan (CS, deacetylation degree of ≥95%, a viscosity of 100–200 mPa.s at 20 °C) was from Aladdin Co. (Shanghai, China). Alginate (SA) was from Macklin Co. (Beijing, China). Vascular endothelial cadherin (VE-cadherin) was purchased from Boster Co. (Wuhan, China). 1-3-Dimethylaminopropyl)-3-ethylarbidiimide Hydrochloride (C_8_H_17_N_3_·HCl, >98.0%) were from TCI Co. (Shanghai, China). N-Hydroxysuccinimide (≥97.0%), Poly(ethylene glycol) diacrylate(PEGDA, Mn ≈ 700 Da) and Rhodamine 123 were obtained from Sigma-Aldrich. The cells’ culture mediums, DMEM-High Glucose, came from BD Biosciences. Servicebio Technology CO., LTD (Wuhan, China) offered hematoxylin-eosin dye. All the other reagents were the highest analytical purity (>99.9%).

### 4.2. Preparation of Complex Hydrogels (CS/SA/PEGDDA Gel)

Briefly, 0.2 g CS was dissolved in 10 mL 0.1 M glacial acetic acid, 1 g PEGDA was dissolved in 10 mL 0.01 M phosphate buffered saline (PBS), and 0.2 g SA was dissolved in 10 mL 0.05 M sodium bicarbonate (NaHCO_3_). Subsequently, PEGDA was added to CS and formed a mixing solution. Then, the mixing solution was added to SA with the catalyzed EDC/NHS.

### 4.3. Swelling Ratio and Degradation Ratio

Swelling tests were used to determine the swelling ratio (SR) of the complex hydrogels, which were analyzed by the weight ratio of dried state (Wd) and fully swollen state (Ws). Firstly, the complex hydrogels were immersed in 0.01 M PBS at 37 °C until the weight of all hydrogels was kept constant. Then the hydrogels were taken out, the superficial water of gels were removed by filter paper, and the weight of the gels was signed as Ws. After that, the gels were dried to a constant weight at 60 °C, and the weight was denoted as Wd. SR was calculated using the following formula: SR = (Ws − Wd)/Wd. The test was repeated three times at least.

Degradation tests were used to determine the degradation ratio (DR) of the complex hydrogels. Firstly, the complex hydrogels were immersed in 0.01 M PBS at 37 °C with shaking at 70 rpm until the weight of all hydrogels was kept constant and the weight were denoted as W_0_. At the predetermined time, the hydrogels were taken out, rinsed using RO water to remove excess salinity, the superficial water of gels were removed by filter paper, and the weight was denoted as W′. DR of hydrogels was analyzed by the following formula: DR = (W_0_ − W′)/W_0_ × 100%. The test was repeated more than three times.

### 4.4. Rheological Properties

Hydrogel samples were prepared into parallel plates (~1 mm high × 25 mm in diameter) for compression test at 25 °C. The Storage Modulus (G′) and Loss Modulus (G″) were carried out using a rheometer (Model DHR-1, TA Instruments) with an oscillation frequency ranging from 0.1 to 100 Hz. All these tests were repeated at least three times.

### 4.5. Scanning Electron Microscope (SEM)

The internal morphology of the complex hydrogels was observed by a scanning electron microscope (SEM). Firstly, the prepared gel was fixed using 2.5% glutaraldehyde for 30 min at 25 °C. Secondly, the hydrogels were put into the liquid nitrogen for −196 °C overnight. Then, the frozen hydrogel was dried using a lyophilizer until all the water within the gels were removed. After that, the gel was cut, and the cross section was fixed on the clean conductive substrate using liquid silver paint to increase the conductivity, and the internal morphology was imaged by SEM (Philips Quanta 2000).

### 4.6. Hemolysis and Whole Blood Dynamic Coagulation Evaluation

The hemolysis test is a necessary experiment for blood compatibility evaluation, which could identify whether the red blood cells are ruptured or not after the hydrogel is connected to the blood. Firstly, the fresh rabbit blood with anticoagulation was centrifuged at 3000 rpm for 15 min to acquire the lower blood cells, and then the blood cells were diluted to 2% using normal saline. The fresh rabbit blood diluted with RO was set as a positive control, and the fresh rabbit blood diluted with normal saline was set as a negative control. Secondly, 1 mL diluted blood cells were added to each hydrogel wells and cultured in an incubator shaker at 37 °C for 1 h. Then the immersion solution was aspirated and centrifuged at 3000 rpm for 5 min to isolate the ruptured blood cells, and the supernate was suctioned to measure the absorbance at 540 nm. The absorbance of hydrogels was signed as A, the absorbance of positive control was signed as B, and the absorbance of negative control was signed as C. The hemolysis test of hydrogels was analyzed by the following formula: (A − C)/(B − C) × 100%.

To evaluate the coagulation effect of the complex hydrogels, the whole blood dynamic coagulation assay in vitro was carried out. Firstly, 150 μL fresh whole blood without anticoagulation was added onto the hydrogel surfaces and incubated at 37 °C for 10, 20, 30, 40 and 50 min, respectively. Secondly, each hydrogel wells were added to RO to dilute the red blood cells that un-involved coagulation. After 5 min, the immersion solution was aspirated and measured the absorbance at 540 nm, and the absorbance of hydrogels was signed as As. Then the hydrogels with sludged blood at different points in time were taken out and photographed. At the same time, 150 μL fresh whole blood with anticoagulation was diluted with RO, measured the absorbance at 540 nm and signed as Aw, which involved the whole red blood cells. Therefore, the blood coagulant index (BCI) of hydrogels were analyzed by the following formula: As/Aw × 100%.

### 4.7. In Vitro and In Vivo Antibacterial Activities

The antibacterial activities of complex hydrogels against Gram-negative bacteria *E. coli* and Gram-positive bacteria *S. aureus* in vitro were evaluated by hydrogel surface antibacterial assay. Firstly, a single colony was seeded in an aseptic beef extract-peptone medium and cultured in an incubator shaker at 37 °C under the speed of 120 rpm for 12 h. Then the bacterial suspension was diluted using sterilized PBS to different bacterial concentration (10^3^, 10^4^, 10^5^, 10^6^, 10^7^,10^8^ CFU/mL). Furthermore, the complex gels were made in culture dishes and placed in UV light for 30 min; LB agar gel plates without hydrogels were as control, and were marked as blank. After that, sterilized PBS was added to the gels to immerse and rinse the gels, and then 10 μL of the bacterial suspension were evenly spaced with different bacterial concentrations on the hydrogel surface and incubated at 37 °C for 12 h. Then the hydrogel plates and agar gel plates were taken photos to observe the colony-forming units (CFUs) on each plate. Then, to detect the CFUs on each plate, 1 mL sterilized PBS was added to each hydrogel plate to dissolve survived bacteria. After that, 10 μL suspension above were added to the surface of agar gel and incubated at 37 °C for 24 h, and 10 μL bacteria as control. After incubation, the number of CFUs on the LB agar gel plates were counted. The killing efficiency was calculated using the following formula: (N_1_ − N_2_)/N_1_ × 100%, where N_1_ refers to the number of CFUs of control, and N_2_ refers to the number of CFUs of the survive to count on complex hydrogels. The antibacterial assay was repeated at least three times.

In this work, all of the experiments with animals were performed with the approval of the Local Ethical Committee of Southwest Jiaotong University and Laboratory Animal Administration Rules of China. All the animals were purchased from Dashuo Experiment Animals CO., LTD (Chengdu, China). Their experimental animal production license was authorized by Sichuan Animal Management Committee, the project identification approval code was “SCXK (chuan) 2020-030”, and date was 6 March 2020. The antibacterial activities of complex hydrogels against Gram-negative bacteria *E. coli* and Gram-positive bacteria *S. aureus* in vivo were evaluated by an antibacterial assay using rats’ full-thickness infected skin defect model. These experiments were carried out using Sprague-Dawley male rats, 3–4 weeks old, weight about 200 g. Firstly, all rats back were shaved and 1.0 cm in diameter wounds were created. Secondly, 1 mL *E. coli* and *S. aureus* of the bacterial suspension (10^8^ CFU/mL) were evenly spaced on the rats’ wounds and maintained for 30 min, respectively. Then the complex hydrogels were added to the infected wounds directly. After the wounds healing for 24 h, those tissues were harvested and fixed using 4% paraformaldehyde overnight. Furthermore, embedding, paraffin section and gram staining to perform histological analysis.

### 4.8. In Vitro Compatibility Evaluation of Epithelial Cells

In order to study the repair effect of complex hydrogel on epithelial tissue, the compatibility of the hydrogels with L929 and EC were evaluated. The EC (EC304) and the L929 were purchased from West Chain Hospital (Chengdu, Sichuan). They were cultured using the medium (DMEM High Glucose) containing 10% fetal bovine serum (FBS) [48].

Before cell experiments, the complex gels were made in 12 pore plates and placed in UV light for 30 min to form a sterile environment. Every type of cells (L929, EC) (3 × 10^4^ cells/mL) were seeded onto the gels’ surface and incubated at 37 °C with 5% CO_2_ for 1,3,7 and 12 days. Besides single cells culture, two type of cells (L929&EC) were also seeded onto the gels. After seeding one type of cell onto the gels for 3 h, another type of cell was seeded. After that, at the pointing time, 630 μL cell medium and 70 μL cell counting kit-8(CCK-8) were added to each sample to investigate the whole sample’s cell number, which includes the surface and the interior cells of the hydrogel. Then, the samples were fixed using 2.5% glutaraldehyde overnight and stained by Rhodamine 123 and DAPI to observe the distribution of cells in the hydrogels using a fluorescence microscope and confocal microscopy (NIKON, A1 + N-ATORM).

### 4.9. In Vivo Wound Healing and Histological Analysis

The wound healing ability of complex hydrogels was evaluated using a full-thickness skin wound model. All the animals were purchased from Dashuo Experiment Animals CO., LTD (Chengdu, China). Their experimental animal production license was authorized by Sichuan Animal Management Committee, the project identification approval code was “SCXK (chuan) 2020-030”, and date was 6 March 2020. Male Sprague-Dawley rats (100–120 g) were anaesthetized, dorsum was shaved, and four round full-thickness wounds (1.0 cm in diameter) were created. Then the wounds were covered with the complex hydrogel disks, and the sterilized PBS were set as control. A gauze was used to cover the wound dress. Ultimately, after wound healing for 3, 7 and 14 days, the skin tissues were excised and fixed using 4% paraformaldehyde overnight. Furthermore, embedding, paraffin section and hematoxylin and eosin (H&E) staining to perform histological analysis.

### 4.10. Statistical Methods

At least three parallel samples were prepared in each experiment. The statistical analysis used Origin 9.0, and data were expressed as mean ± standard deviation (SD). And one-way ANOVA tests were employed to show statistical differences between groups, * *p* < 0.05 was considered statistically significant.

## 5. Conclusions

A series of hemostatic, antibacterial and super cytocompatibility complex hydrogels based on PEGDA, CS and SA, were successfully prepared through the crosslinking of the carboxyl of alginate and the amino and the hydroxy of chitosan. These complex hydrogels exhibited multifunction like suitable swelling, tunable pore size and rheological property and controlled degradation behaviors. Besides, the presents of chitosan endowed hydrogels with excellent hemostatic and antibacterial ability, which solved the problem of hemostasis and inflammation at the wound healing first two stages. Besides, the basic hydrogels could promote the adhesion and proliferation of L929 and EC to a certain degree. Additionally, the C8-S2 hydrogel with FGF and VE-cadherin exhibited unexceptionable cytocompatibility to the co-culture of L929 and EC, especially in the proliferation of the two cells after culturing for 7 and 12 days. The hydrogels could provide a 3D cell culture niche to enhance cell–cell and cell–ECM links, and the introduction of FGF and VE-cadherin could promote the proliferation and migration of cells through the PI3K/AKT pathway signaling pathways. Furthermore, the histological evaluation showed a better wound healing effect in a full-thickness skin defected model. These properties make the complex hydrogels more competitive in the promising application of a full-thickness skin wound healing process.

## Figures and Tables

**Figure 1 ijms-23-01249-f001:**
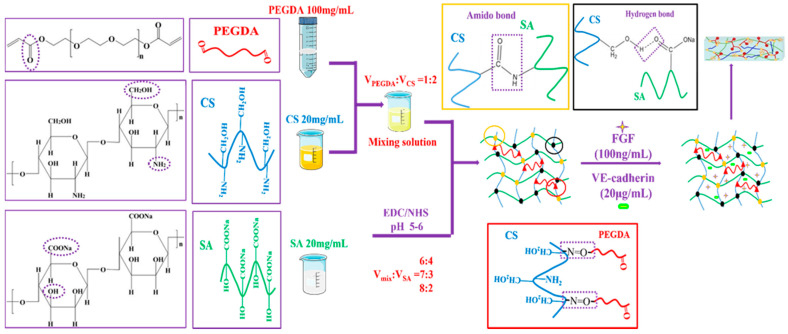
The structural formula of PEGDA, CS, SA and the schematic drawing of complex hydrogels.

**Figure 2 ijms-23-01249-f002:**
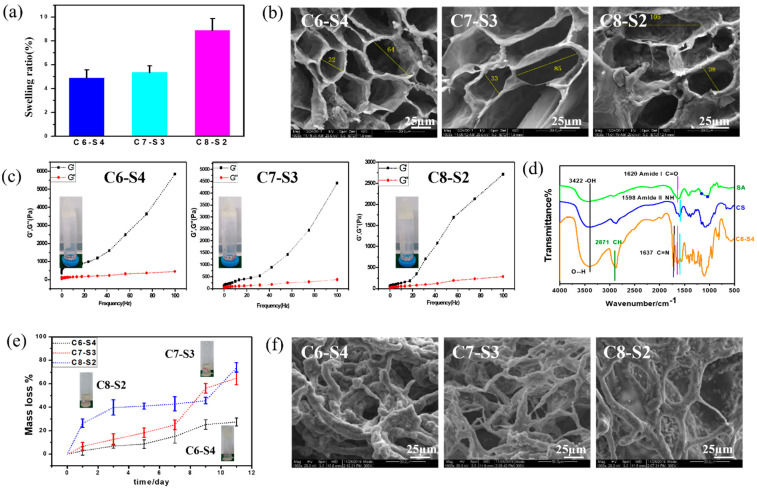
Physical characterization of complex hydrogels. (**a**) The swelling ratio of complex hydrogels (mean ± SD, *n* ≥ 3). (**b**) SEM micrograph of complex hydrogels. The scale bar stands for 25 µm in (**b**). (**c**) Rheological behavior of complex hydrogels, in which the forming state of com- plex hydrogels were showed in the left of each picture. (**d**) The FTIR of alginate (SA), chi-tosan (CS) and C6-S4 hydrogel. (**e**) The mass loss of complex hydrogels within 11 days of degra-dation. (**f**) SEM micrograph of complex hydrogels after 11 days of degradation. The scale bar stands for 25 µm in (**f**).

**Figure 3 ijms-23-01249-f003:**
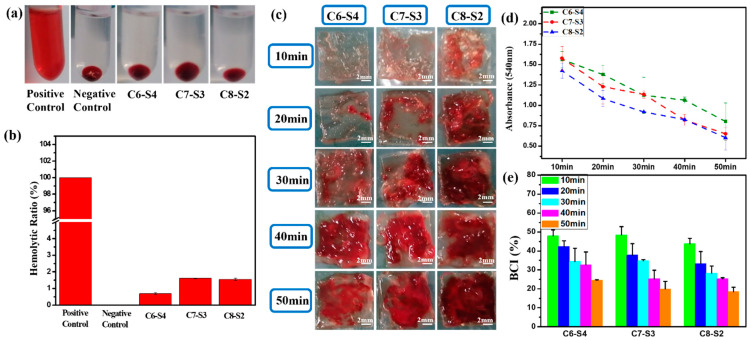
Blood compatibility evaluation of complex hydrogels. (**a**) The photos of complex hydrogels). (**b**) The hemolytic ratio of complex hydrogels. (mean ± SD, *n* ≥ 3). (**c**) The photos of whole blood assay of complex hydrogels at different times. The scale bar stands for 2mm in (**c**). (**d**) The whole blood assay absorbance of complex hydrogels at different times. (mean ± SD, *n* ≥ 3). (**e**) The BCI of complex hydrogels at different times (mean ± SD, *n* ≥ 3).

**Figure 4 ijms-23-01249-f004:**
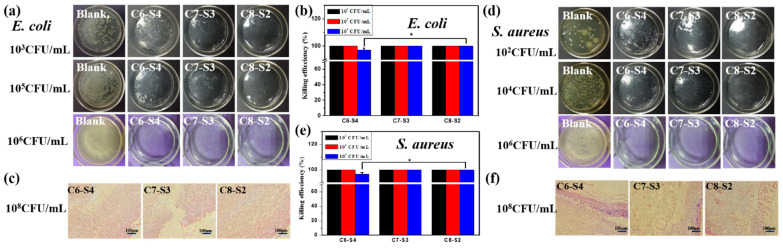
Antibacterial activity evaluation of complex hydrogels. (**a**) Bacteriostatic pictures of complex hydrogels for *E. coli* under the 10^3^ CFU/mL, 10^5^ CFU/mL and 10^6^ CFU/mL bacterial. (**b**) The killing efficiency of hydrogels against *E. coli*. (mean ± SD, *n* ≥ 3, * *p* < 0.05). (**c**) Gram stain pictures of complex hydrogels after animals affected with 10^8^ CFU/mL *E. coli* for 24 h. The scale bar stands for 100 µm in (**c**). (**d**) Bacteriostatic pictures of complex hydrogels for *S. aureus* under the 102 CFU/mL, 104 CFU/mL and 106 CFU/mL bacterial concentration. (**e**) The killing efficiency of hydrogels against *S. aureus*. (mean ± SD, *n* ≥ 3, * *p* < 0.05). (**f**) Gram stain pictures of complex hydrogels after animals affected with 108 CFU/mL *S. aureus* for 24 h. The scale bar stands for 100 µm in (**f**).

**Figure 5 ijms-23-01249-f005:**
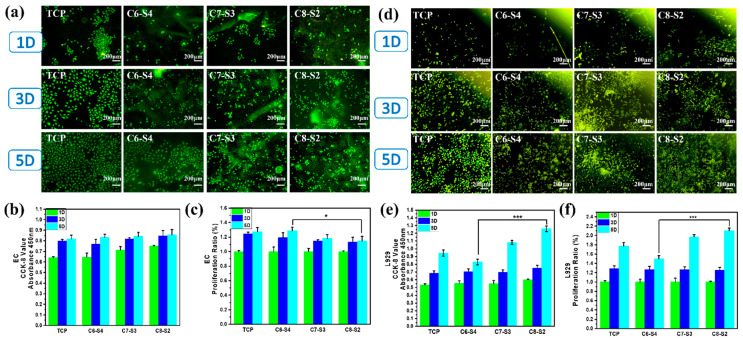
Evaluation of cells compatibility. (**a**) EC morphology of complex hydrogels. The scale bar stands for 200 µm in (**a**). (**b**) The CCK-8 results of EC. (mean ± SD, *n* ≥ 3, * *p* < 0.05). (**c**) The proliferation ratio of EC. (mean ± SD, *n* ≥ 3, * *p* < 0.05). (**d**) L929 morphology of complex hydrogels. The scale bar stands for 200 µm in (**d**). (**e**) The CCK-8 results of L929. (mean ± SD, *n* ≥ 3, *** *p* < 0.01). (**f**) The proliferation ratio of l929. (mean ± SD, *n* ≥ 3, *** *p* < 0.01).

**Figure 6 ijms-23-01249-f006:**
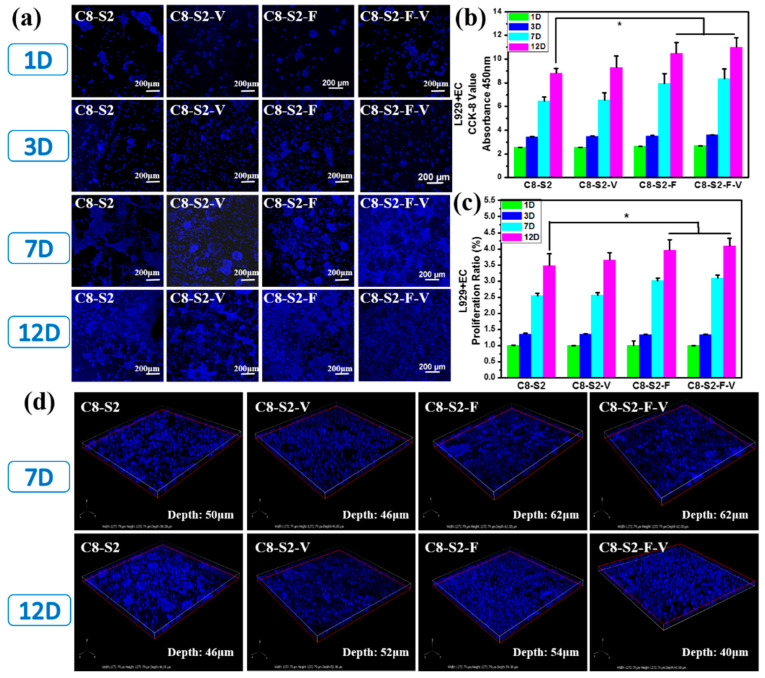
Evaluation of L929&EC co-culture compatibility. (**a**) Morphology of L929&EC of complex hydrogels. The scale bar stands for 200 µm in (**a**). (**b**) the CCK-8 results of L929&EC; (**c**) the proliferation ratio of L929&EC. (mean ± SD, *n* ≥ 3, * *p* < 0.05); (**d**) the 3D morphology of L929&EC and the complex hydrogels.

**Figure 7 ijms-23-01249-f007:**
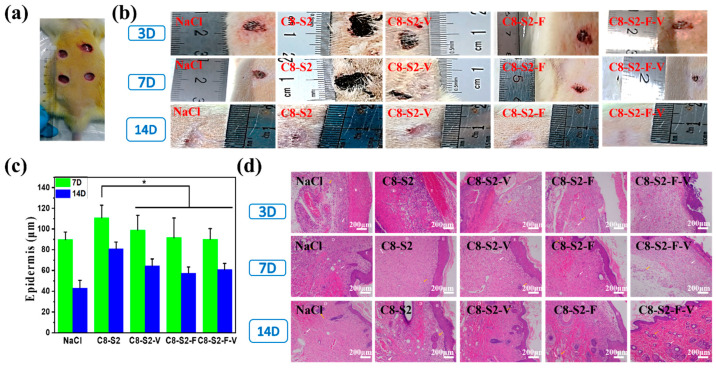
In vivo repair and regeneration evaluation of complex hydrogels. (**a**) Skin repair trauma experiments of complex hydrogels; (**b**) The appearance of skin wound for 3, 7, 14 days observation of repair experiments; (**c**) Epidermis thickness for different groups on 7th and 14th days. (mean ± SD, *n* ≥ 3, * *p* < 0.05) (**d**) H&E staining of tissue sections of skin wound for 3, 7, 14 days (blood vessels: white arrows; hair follicles: orange arrows; E were epidermal layer and D were dermal layer). The scale bar stands for 200 µm in (**d**).

**Figure 8 ijms-23-01249-f008:**
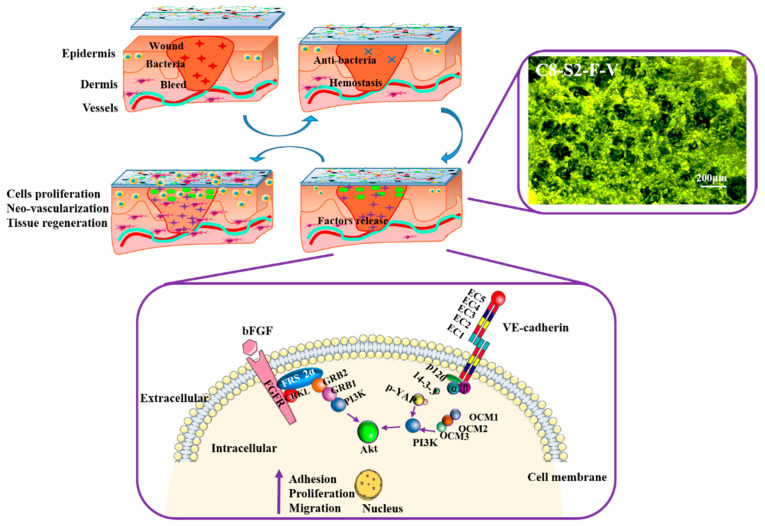
The relationship between hydrogels and wound healing process, involved in a cellular fluorescence image of the C8-S2-F-V hydrogel and the signaling pathways of FGF and VE-cadherin; CCM1, cerebral cavernous malformation protein 1; PI3K, phosphatidylinositol 3 kinases; YAP, yes-associated protein; FGFR, FGF receptors. The scale bar stands for 200 µm in Figure 8.

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
