# Peer review of "Chitosan/Alginate Hydrogel Dressing Loaded FGF/VE-Cadherin to Accelerate Full-Thickness Skin Regeneration and More Normal Skin Repairs"

_ijms, 2022, doi:10.3390/ijms23031249_

Round 1

Reviewer 1 Report

The manuscript under evaluation in International Journal Molecular Sciences, entitled “Chitosan/alginate hydrogel dressing loaded FGF/VE-cadherin: 2 To accelerate full-thickness skin regeneration and more normal 3 skin repairs” are an interesting manuscript concerning the use of loaded hydrogels with FGF/VE-cadherin for skin regeneration. 

To my opinion, this is a manuscript about an interesting topic with important points to be corrected and clarified. Please check the English all over the manuscript, correct some words.

In the abstract, Line 24, the authors mentioned “hydrogels with growth factors have tremendous potential as multifunctional.” to our opinion “tremendous” could be replaced to a different word.

Line 57. Please correct “hydrophilcity”

Line 77-78. To our opinion the authors must improve about their statement related to “FGF enhancing angiogenesis and so on”

Line 89. The authors should introduce the meaning of PEGDA, as it is the first time it appears in the text, as well as in

Line 295: the meaning of EDC/NHS

Line 95. The authors mentioned “…were carried out to evaluate the excellent effects of full-thickness wound healing”, to our opinion the goal was to evaluate the effects of full thickness wound healing, “excellent” was a conclusion.

Line 106. To our opinion the authors must indicate the meaning of letters in the equation: SR= (Ws-Wd)/Wd, Line 106: as well as in equation DR= (W0- W′)/ W0×100%.

Line 222-223. To our opinion the following sentence must be revised “To further evaluate the cytocompatibility of hydrogels, the hydrogel encapsulated 222 the L929 cells and ECs were carried out”

Line 306. In the sentence “…same brilliant antibacterial effect.”, “brilliant” may not be considered the best scientific word.

Line 309. The authors mentioned that  “There were pieces of  literature that showed that chitosan could induce platelet adhesion and aggregation” this sentence must be reformulated and the respective references added.

Line 311. In this sentence at least one reference must be added:  “Recently, more evidence showed that the hemostatic pro-311 motion of chitosan was independent of traditional clotting pathways”

Line 319. Please correct “bacterial” to “bacteria”.

Line 367. Please correct “mpa.s” to “mPa.s”

Line 447-448. Please correct “beef extract-peptone medium” to “aseptic beef extract-peptone medium”

Author Response

  1. In the abstract, Line 24, the authors mentioned “hydrogels with growth factors have tremendous potential as multifunctional.” to our opinion “tremendous” could be replaced to a different word.

Re: “tremendous” have been changed to “great”, as showed in manuscript.

  1. Line 57. Please correct “hydrophilcity”

Re: “hydrophilcity” have been changed to “hydrophilicity” , as showed in manuscript.

  1. Line 77-78. To our opinion the authors must improve about their statement related to “FGF enhancing angiogenesis and so on”

Re: This sentence was changed into “FGF could also enhancing angiogenesis which promote the formation of new vessels from pre-existing vessels”, were added and numbered 29-31.

  1. Prudovsky, I., Cellular Mechanisms of FGF-Stimulated Tissue Repair. Cells 2021, 10(7), 1830.
  2. Xie, Y.; Su, N.; Yang, J.; Tan, Q.; Huang, S.; Jin, M.; Ni, Z.; Zhang, B.; Zhang, D.; Luo, F.; Chen, H.; Sun, X.; Feng, J. Q.; Qi, H.; Chen, L., FGF/FGFR signaling in health and disease. Signal Transduct Target Ther 2020, 5(1), 181.
  3. Mossahebi-Mohammadi, M.; Quan, M.; Zhang, J. S.; Li, X., FGF Signaling Pathway: A Key Regulator of Stem Cell Pluripotency. Front Cell Dev Biol 2020, 8, 79.

  1. Line 89. The authors should introduce the meaning of PEGDA, as it is the first time it appears in the text, as well as in

Re: This sentence was changed into “Poly(ethylene glycol) diacrylate (PEGDA)”, as showed in manuscript.

  1. Line 295: the meaning of EDC/NHS

Re: This sentence was changed into “1-3-Dimethylaminopropyl-3-ethylarbidiimide Hydrochloride (EDC, C8H17N3·HCl) and N-Hydroxysuccinimide (NHS) is often used as binding agents of the amino and carboxyl groups”, as showed in manuscript.

  1. Line 95. The authors mentioned “…were carried out to evaluate the excellent effects of full-thickness wound healing”, to our opinion the goal was to evaluate the effects of full thickness wound healing, “excellent” was a conclusion.

Re: This sentence was changed into “were carried out to evaluate the effects of full-thickness wound healing”, as showed in manuscript.

  1. Line 106. To our opinion the authors must indicate the meaning of letters in the equation: SR= (Ws-Wd)/Wd, Line 106: as well as in equation DR= (W0- W′)/ W0×100%.

Re:

Swelling tests were used to determine the swelling ratio (SR) of the complex hydrogels, which were analyzed by the weight ratio of dried state (Wd) and fully swollen state (Ws). The equilibrium mass swelling of complex hydrogel was calculated according to the formula: SR= (Ws-Wd)/Wd. It was showed in paragraph 2 of 2.1 in manuscript.

Degradation tests were used to determine the Degradation ratio (DR) of the complex hydrogels. Firstly, the complex hydrogels were immersed in 0.01M PBS at 37℃with shaking at 70 rpm until the weight of all hydrogels was kept constant and the weight were denoted as W0. At the predetermined time, the hydrogels were taken out, rinsed using RO water to remove excess salinity, the superficial water of gels were removed by filter paper, and the weight was denoted as W′. DR of hydrogels was analyzed by the formula: DR= (W0- W′)/ W0×100%. It was showed in paragraph 6 of 2.1 in manuscript.

  1. Line 222-223. To our opinion the following sentence must be revised “To further evaluate the cytocompatibility of hydrogels, the hydrogel encapsulated 222 the L929 cells and ECs were carried out”

Re: This sentence was changed into “To further evaluate the cytocompatibility of hydrogels, the L929 cells and ECs were encapsulated into the hydrogels”, as showed in manuscript.

  1. Line 306. In the sentence “…same brilliant antibacterial effect.”, “brilliant” may not be considered the best scientific word.

Re: This sentence was changed into “And all the hydrogels own the same excellent antibacterial effect.”, as showed in manuscript.

  1. Line 309. The authors mentioned that “There were pieces of literature that showed that chitosan could induce platelet adhesion and aggregation” this sentence must be reformulated and the respective references added.

Re: This sentence was changed into “There were lots of literatures have showed that chitosan could induce platelet adhesion and aggregation.” The reference “Chou T C, Fu E, Wu C J, et al. Chitosan enhances platelet adhesion and aggregation. Biochemical and Biophysical Research Communications, 2003, 302(3), 480-483.” was added and numbered 37. 

  1. Line 311. In this sentence at least one reference must be added: “Recently, more evidence showed that the hemostatic pro-311 motion of chitosan was independent of traditional clotting pathways”

Re: The reference “Rondon EP, Benabdoun HA, Vallières F, et al. Evidence Supporting the Safety of Pegylated DiethylaminoethylChitosan Polymer as a Nanovector for Gene Therapy Applications. Int J Nanomedicine 2020, 15, 6183-200” was added and numbered 39.

  1. Line 319. Please correct “bacterial” to “bacteria”.

Re: “bacterial” has been changed into “bacteria”, as showed in manuscript.

  1. Line 367. Please correct “mpa.s” to “mPa.s”

Re: “mpa.s” has been changed into “mPa.s”, as showed in manuscript.

  1. Line 447-448. Please correct “beef extract-peptone medium” to “aseptic beef extract-peptone medium”

Re: “beef extract-peptone medium” has been changed into “aseptic beef extract-peptone medium”, as showed in manuscript.

Reviewer 2 Report

1- The Fig. 1 should be mentioned and cited properly in the main text.

2- CxSy abbreviations should be defined at the proper position

3- in page 11, Experimental: Chitosan is a powder. What do you mean by "viscosity"?

4- No evidence of the formation of the Cs-Alg hydrogel has been given, e.g.  a comparison of FTIR of all the components and the products.

5- Page 9, Discussion. There is no evidence of the discussed chemical bonds.

6- In line 225: the blank has been defined as tissue culture plate (blank). Is it the same as in the rest of the paper, e.g. in Fig. 7c??

7- A comparison of the effectiveness of the prepared hydrogels with pure Cs and/or pure alginate should be given for all the investigations.

Round 2

Reviewer 2 Report

The suggested corrections have been made properly.

Author Response

  1. The spell has been checked, and the changes have been highlighted.  Please see the attachment.
